# An Indoor Autonomous Inspection and Firefighting Robot Based on SLAM and Flame Image Recognition

Sen Li [1], Junying Yun [1], Chunyong Feng [2], Yijin Gao [1], Jialuo Yang [1], Guangchao Sun [1] and Dan Zhang [1,*]

1   The School of Building Environment Engineering, Zhengzhou University of Light Industry,
    Zhengzhou 450001, China
2   The School of Mechanical and Electrical Engineering, Xi'an University of Architecture and Technology,
    Xi'an 710055, China
*   Correspondence: zhangdan@zzuli.edu.cn

**Abstract:** Indoor fire accidents have become increasingly common in recent years. More and more firefighting robots have been designed to solve the problem of fires. However, the indoor environment is very complex, with high temperatures, thick smoke, more turns, and various burning substances. In this study, a firefighting robot with autonomous inspection and automatic fire-extinguishing functions intended for use in unknown indoor environments was designed. Considering water's poor efficiency and its inability to extinguish some combustion materials, other fire extinguishers were applied to design the extinguishing system. The robot can install four different extinguishers as required and select the appropriate fire extinguisher to spray it automatically. Based on the Cartographer SLAM (simultaneous localization and mapping) theory, a robot map-building system was built using Lidar scanners, IMU (inertial measurement unit) sensors, encoders, and other sensors. The accurate identification and location of the fire source were achieved using an infrared thermal imager and the YOLOv4 deep learning algorithm. Finally, the performance of the firefighting robot was evaluated by creating a simulated-fire experimental environment. In an autonomous inspection process of the on-fire environment, the firefighting robot could identify the flame in real-time, trigger the fire-extinguishing system to carry out automatic fire extinguishing, and contain the fire in its embryonic stage.

**Keywords:** firefighting robot; laser SLAM; YOLOv4 deep learning algorithm; autonomous inspection; automatic fire extinguishing



## 1. Introduction

More and more people live and work in dense indoor buildings with the acceleration of urbanization [1]. Large numbers of casualties and serious economic losses can be caused when indoor building fires occur. From a global perspective, the cost of fires is high. According to statistics, the direct and indirect losses caused by fires account to ~1% of global GDP [2]. Public fire departments across the U.S. attended 499,000 fires in indoor buildings in 2018, which caused 2910 deaths and 12,700 injuries [3]. The *Chinese Fire Statistical Yearbook* recorded 395,052 indoor building fires in China, resulting in 1815 deaths, 1513 injuries, and a loss of RMB 4.7 billion in 2014 alone [4]. Why are there so many serious indoor building fires? On the one hand, a fire accident occurs because the fire was not detected promptly, resulting in the fire's rapid spread. On the other hand, firefighting facilities are inadequate, and effective firefighting cannot be carried out once a fire has been discovered [5,6]. Some practical measures to reduce the frequency of major fires are to learn to locate the fire source early in the fire, quickly and accurately mark the fire source, and automatically carry out fire extinguishing. Given the above situation, the development of firefighting robots with autonomous inspection and automatic extinguishing functions has evolved into a new development direction in the field of fire protection.

A firefighting robot can assist or replace firefighters in carrying out firefighting work, reducing firefighters' labor intensity and improving their firefighting and rescue ability and efficiency. Scholars worldwide have been working on developing firefighting robots as robotics technology advances. Liu [7] designed a firefighting robot with multiple sensor fusions. The robot precisely positions itself in smoke scenes by fusing magnetic encoder trajectory estimation poses, UWB absolute poses, and IMU heading angle poses through an improved Kalman filtering algorithm. The shortcoming of this design is that the robot needs to be designed with a fire extinguishing system. Yazhou Jia [8] developed a crawler firefighting robot. It is equipped with a water gun device on the front end of the robot's body and a water pipe on the back end, and it is equipped to accurately detect high-temperature objects in a fire scene and assist firefighters in firefighting. Xinyu Gao [9] developed a firefighting robot capable of autonomous navigation and extinguishing. The front part of the robot is equipped with a water cannon. The water cannon must use an external fire hydrant to extinguish distant flames. SnakeFighter [10] is a snake-like hydraulic robot. The robot has a function that allows it to avoid obstacles while walking. The front-mounted nozzles on the robot spray onto the fire.

The firefighting robot LUF60 [11] has a blower and a water mist nozzle. The blower refines the water flow and sprays it as water mist, reducing the ambient temperature over a wide range to achieve the goal of firefighting. The robot's back end is connected to a water hose so that it can spray water mist indefinitely. The LUF60 firefighting robot is shown in Figure 1. The firefighting robot GARM [12] has a tele-operated monitor nozzle and an IR video system that can detect hidden glows and other heat sources. The robot can often automatically detect and extinguish remaining glows and newly igniting fires. The firefighting robot Thermite 3.0 [13] has a multidirectional nozzle that can pump 2500 gallons of water per minute from the cannon. The robot can continuously spray water that is pumped in through the rear hose. Shuo Zhang [14] designed an intelligent firefighting robot based on multisensor fusion technology with autonomous inspection and automatic firefighting functions. The robot uses the tandem operation of a binocular vision camera and an infrared thermal imager to locate a fire source. Like the above robot, this robot uses water to extinguish fires.

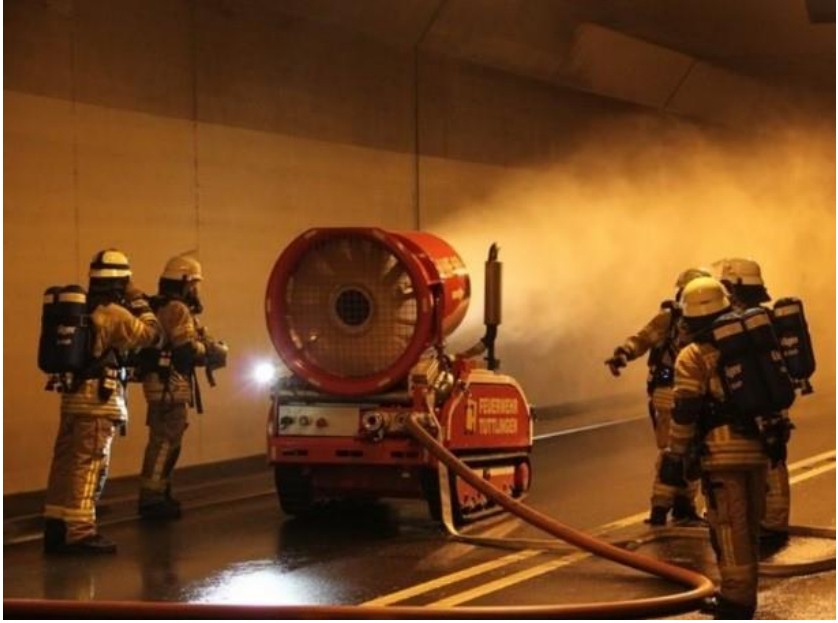

**Figure 1.** LUF60 robot.

In addition, there are the MVF-5 [15], the WATER CANNON [16], aerial-hose-type robots that use water jets [17], intelligent firefighting robots [18], elevating firefighting

robots Based on ROS elevating firefighting robots [19], and others. These firefighting robots are currently unsuitable for use in indoor environments, as evidenced by the following factors: (1) Because of the large size of these firefighting robots, it is inconvenient for them to move indoors. (2) Currently, these firefighting robots use water as the primary extinguishing medium and require an external water hose, but the indoor space is often small, there are many turns, and the water hose restricts the robots' movement. (3) Water as a medium has a low fire-extinguishing efficiency and cannot meet the fire-extinguishing needs of various burning substances.

This paper discusses the development of a firefighting robot for indoor use based on the abovementioned issues. The robot not only has an autonomous inspection function but also has an automatic fire-extinguishing function. The firefighting robot has a flexible appearance, does not need an external water source, and can move freely around the house. Moreover, the robot can also utilize different fire-extinguishing media according to the different burning materials identified.

## 2. Related Work

Currently, the four primary components of the laser SLAM framework for mobile robots are front-end matching, back-end optimization, loopback detection, and map building [20]. The SLAM framework is shown in Figure 2. According to the different back-end optimization methods, it can be divided into filter-based SLAM and graph-optimization-based SLAM [21]. The first and most popular SLAM approach is filter-based. It is possible to consider filter-based SLAM as a continuous Bayesian estimation process. Extended Kalman filters [22], particle filters [23], and Rao–Blackwellized particle filters [24] are the three basic filter steps that SLAM goes through. Based on the RBPF, Montemerlo et al. [25] proposed the FastSLAM method, while Grisetti et al. [26] proposed the Gmapping SLAM algorithm.

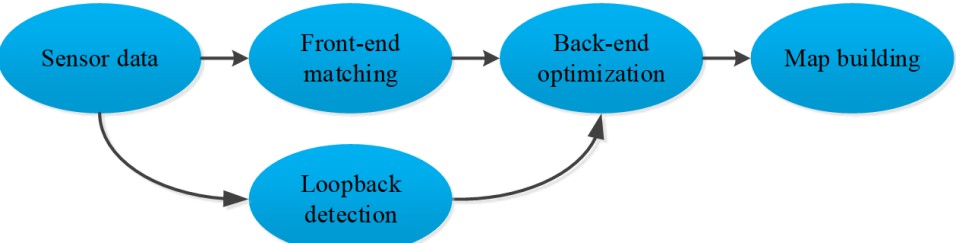

**Figure 2.** SLAM framework.

In SLAM based on graph optimization, the pose of robots at various times is represented as vertices in the graph. The interaction between vertices due to spatial constraints is represented as edges in the graph [27]. The spatial constraint relationship between vertices is obtained by matching odometer information or observation information. Then, the SLAM problem is transformed into a graph optimization problem to solve the optimal robot pose in attitude space under the premise of satisfying constraint relations [28]. The Cartographer SLAM algorithm was proposed by Hess et al. [29] and is based on a standardized graph-optimized SLAM framework. The algorithm consists of two parts: front end and back end. The front end is used to construct the pose map, and the back end is used to optimize the pose map. This algorithm can build a map with good quality in a large-scale environment because the concept of submap is proposed, and the accumulated errors between submaps can be eliminated by loopback detection. In this study, the Cartographer SLAM algorithm was used for robot map construction.

Firefighting robots mostly use visual sensors for flame detection and location, and they carry out firefighting and rescue according to the flame location information [30]. Thermal imaging cameras and visible cameras are two categories of visual sensors. Since visible-light cameras are significantly impacted by smoke, they cannot detect flames in smoke environments. Thermal imaging cameras, which passively pick up the object's thermal radiation, are unaffected by the smoke. Therefore, they can detect flames in a

smoke-filled atmosphere. There are currently three methods for flame detection systems based on vision sensors: traditional image detection, traditional machine learning, and deep learning [31].

Different flame detection systems have been designed by scholars such as Toreyin [32], Celik [33], Marbach [34], and Lin [35] based on traditional image processing methods. Among the flame detection technologies based on traditional machine learning, the most representative are the flame detection methods proposed by Byoung [36], Borges [37], and Chi et al. [38]. The deep-learning-based flame detection technology uses a convolutional neural network to design a self-learning classifier, which can automatically extract flame features. Currently, there are two types of object detectors based on convolutional neural networks: two-stage and one-stage. One-stage algorithms with the highest representation include YOLO, SSD, and RetinaNet [39]. One of the best target detection algorithms that strikes a compromise between speed and accuracy is the YOLOv4 method, which was recently proposed. The YOLOv4 method is used as the flame detection algorithm in this paper. The network structure of the YOLOv4 target detection model is shown in Figure 3.

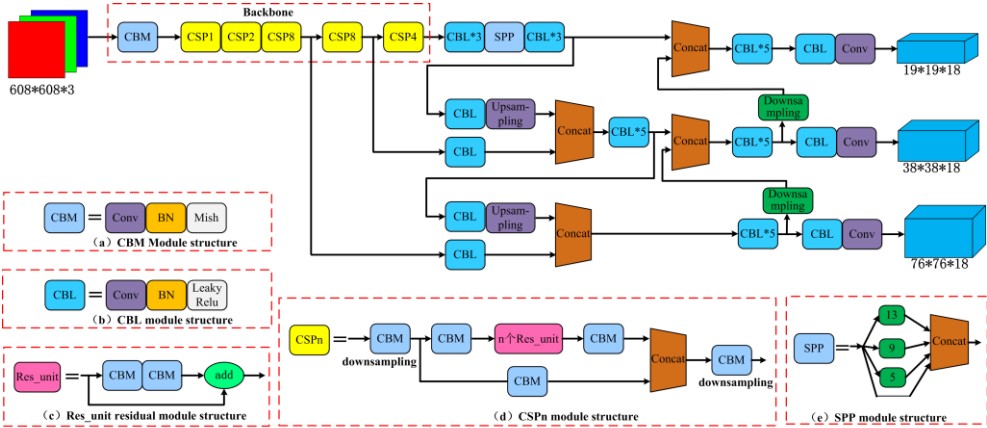

**Figure 3.** The network structure of the YOLOv4 target detection model.

## 3. Hardware Design of the Autonomous Inspection and Firefighting Robot

The hardware design for autonomous inspection and firefighting robots is divided into two sections: autonomous inspection, and automatic firefighting. The autonomous inspection system includes a motion unit and a sensor unit. The motion unit comprises a mobile carrier, a drive controller, an industrial computer, and a power supply system, while the sensing unit comprises a Lidar, an IMU, and an encoder. The automatic fire-extinguishing system includes video monitoring and fire-extinguishing systems. The video monitoring system uses a camera and a pan/tilt to detect and locate the flame. The fire-extinguishing system consists of a fire extinguisher trigger device, a fire-extinguishing system rotating device, and a proximity switch. The hardware system architecture is depicted in Figure 4.

The robot's motion unit controls the robot's motion state. The Lidar scans the environment to generate laser point cloud information, and the encoder and IMU estimate the robot's pose as it moves. The SLAM algorithm in the industrial computer creates its positioning and map based on the laser point cloud information and the robot posture information. Using the positioning and map data, the robot conducts an autonomous inspection. The image data collected by the video surveillance system and the map created by SLAM are communicated to the host computer via the wireless local area network for real-time viewing. At the same time, the flame detection algorithm on the host computer identifies and locates the flame. The fire-extinguishing system targets the flame and activates the fire-extinguishing device to put out the fire, completing the robot's automatic fire-extinguishing function. Figure 5 depicts the autonomous inspection firefighting robot proposed in this paper.

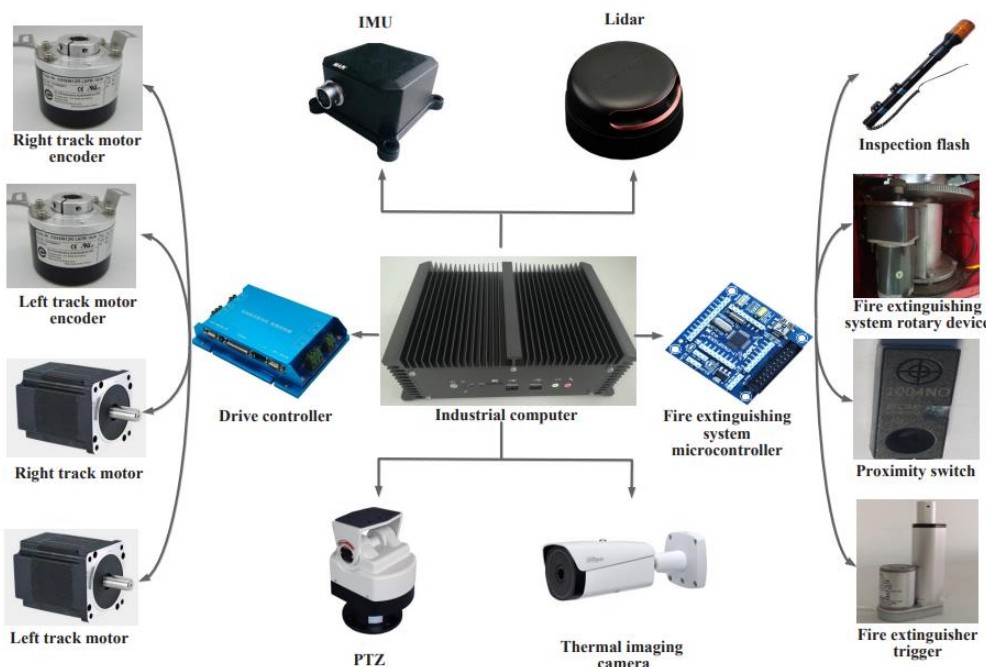

**Figure 4.** Schematic diagram of the robot's hardware system architecture.

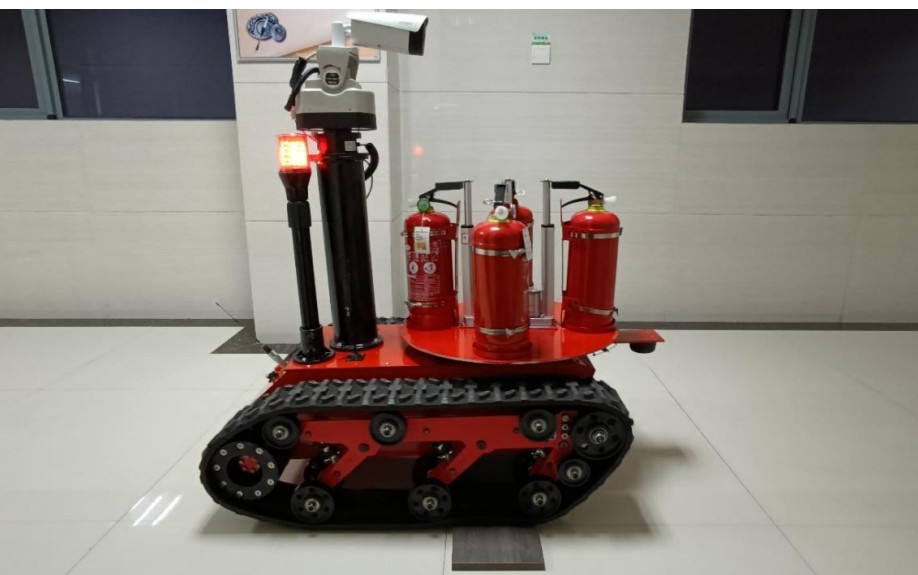

**Figure 5.** Indoor autonomous inspection firefighting robot.

*3.1. Design of the Robot Map Construction System*

3.1.1. Robot Motion Unit

The hardware of the robot motion unit is shown in Figure 6. The robot must possess certain abilities to overcome obstacles in the fire field. The Safari-600T crawler chassis was chosen as the mobile carrier in this study. The crawler support surface is crawler-toothed and has strong traction and adhesion, meeting the criteria for climbing obstacles. The robot moves around by controlling the rotation of the motor. Each of the brushless DC motors on the left and right tracks of the robot chassis was designed in this study. To control two DC brushless motors simultaneously, the drive controller selects the KYDBL4850- 2E controller. The power supply used for the firefighting robot described in this paper must be strong because it must transport a large volume of fire-extinguishing materials. This paper chose a 48 V, 30 AH LiFePO4 battery with a large capacity and high temperature resistance to power the robot.

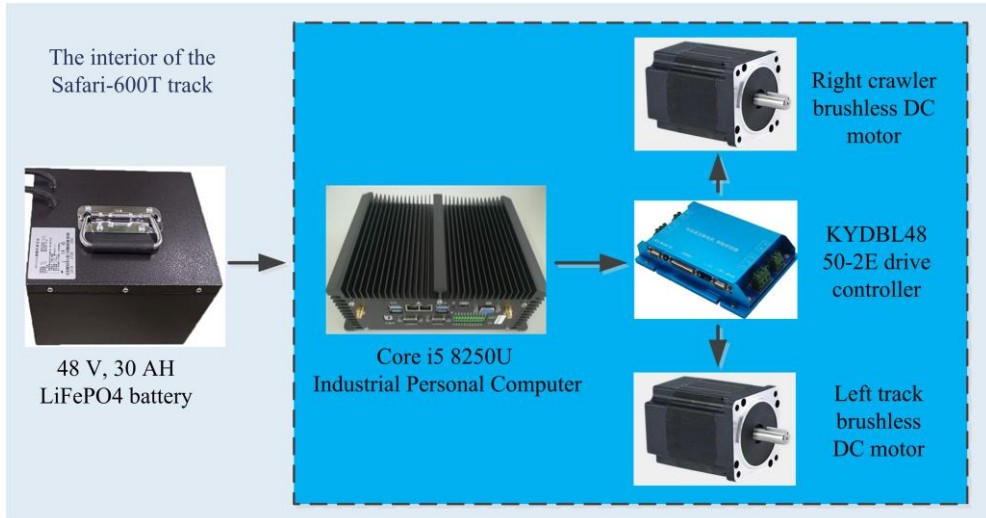

**Figure 6.** The main hardware of the motion unit.

The industrial computer in the robot directs the work of the drive controller. This robot uses a Core i5 8250U industrial computer for robot data processing and algorithm operations, because the Core i5 8250U is a high-performance industrial computer that is dustproof, small in size, and can run stably in a fire field. The industrial computer is housed in the inside cabin of the robot's chassis. The industrial computer's antenna is connected to the robot's shell for the industrial computer to receive information reliably, and the chosen antenna is a 2.4G/Wi-Fi high-gain antenna.

3.1.2. Robot Sensor Unit

The robot sensor unit comprises three sensors: Lidar, IMU, and encoder. The sensor placement positions are schematically illustrated in Figure 7. The RPLIDAR A2 Lidar is positioned upside-down in front of the robot, 0.7 m from its center point (the Lidar technical parameters are shown in Table 1). In the robot's center, the SC-AHRS-100D IMU is mounted. The incremental 1024-line encoder EB58W12R is installed at the crawler drive motor's tail.

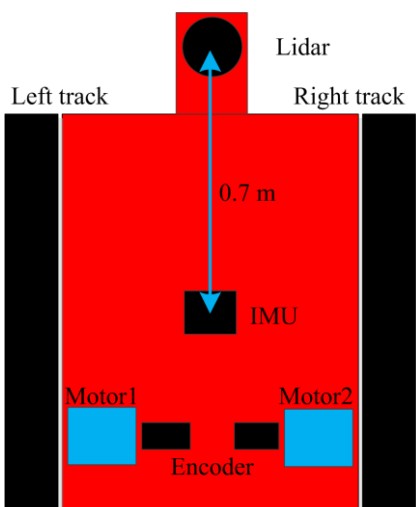

**Figure 7.** Schematic diagram of the sensor installation positions.

**Table 1.** Main parameters of the RPLIDAR A2 Lidar.

| Technical Indicators | Typical Value | Maximum Value |
| --- | --- | --- |
| Ranging range | 0.2~12 m | Not applicable |
| Ranging resolution | <0.5 mm | Not applicable |
| Scan angle | 0~360° | Not applicable |
| Scanning frequency | 10 Hz | 15 Hz |
| Measurement frequency | Magnetization | 8010 Hz |
| Angular resolution | Magnetization | 1.35° |
| Laser wavelength | 785 nm | 795 nm |

The coordinate system of the RPLIDAR A2 Lidar is based on the left-hand rule, whereas the ROS system is based on the right-hand rule. In the left- and right-handed coordinate systems, the direction of a specific coordinate axis is reversed. The forward direction of the robot is set as the positive direction of the *x*-axis, and the *x*-axis of the Lidar is set as the reverse direction to complete the transition of the left- and right-hand coordinate systems. The installation of the robot Lidar is depicted in Figure 8. The robot's primary sensor is the Lidar. The Lidar is installed upside-down to protect it from falling objects or the fire-extinguishing substances sprayed by the fire extinguisher. The conversion coordinate system must be changed in the startup file because the coordinate system of the Lidar is mounted upside-down, and the coordinate systems of the usual installation are mirror images of one another. Due to the robot's shell and track shielding, the Lidar scanning range is 0–270°.

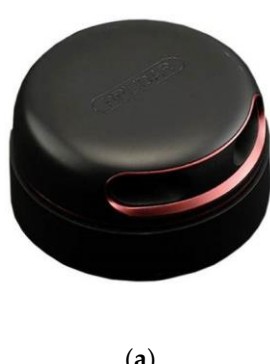

(**a**)

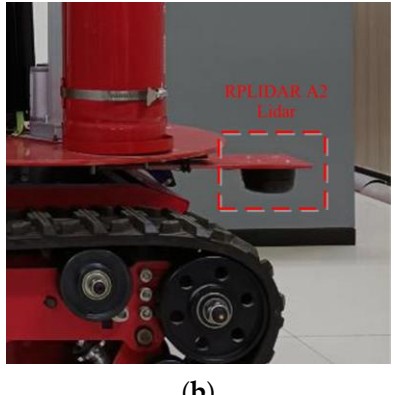

(**b**)

**Figure 8.** Lidar mounted upside-down on the robot: (**a**) Lidar. (**b**) Lidar mounted on the robot.

### 3.1.3. Robot Kinematics Model

The robot's driving mechanism is rear-drive, with an independent driving wheel on each left and right track. The robot's motion node controls the two driving wheels' speed changes, allowing the robot to perform a variety of maneuvers. Using encoder data from the drive wheel motor, the robot's motion state is determined in real time, and the robot's pose and odometer data are then estimated. The current robot's real-time control information is based on the odometer feedback information, which is used to give speed control instructions. The robot's motion control node calculates the speed of the driving wheel motor and converts the motor speed into the robot's forward speed in response to the speed command. The robot's steering is determined by the difference between the left and right motors. The calculation structure of the tracked robot's kinematic solution is integrated into the ROS robot motion control node to achieve motor speed control.

Figure 9 depicts the robot's motion model, where $V$ is the robot's linear velocity, $\omega$ is the robot's angular velocity, $O$ is the robot's center position, $V_l$ is the robot's left track speed, $Vr$ is the robot's right track speed, $R$ is the robot's turning radius, and $L$ is the robot's wheel spacing. The turning radius, angular velocity, and linear velocity of the robot are calculated using Formulas (1)–(3), respectively. The linear and angular velocity of the robot

can be used to calculate the travel speed of the left and right tracks. The motor speed is then calculated based on the track driving wheel's diameter.

$$R = \frac{L(V_r + V_l)}{2(V_r - V_l)}, \tag{1}$$

$$\omega = \frac{V_r - V_l}{L}, \tag{2}$$

$$V = \omega \times R = \frac{V_r - V_l}{L} \times \frac{L(V_l + V_r)}{2(V_r - V_l)} = \frac{V_l + V_r}{2}, \tag{3}$$

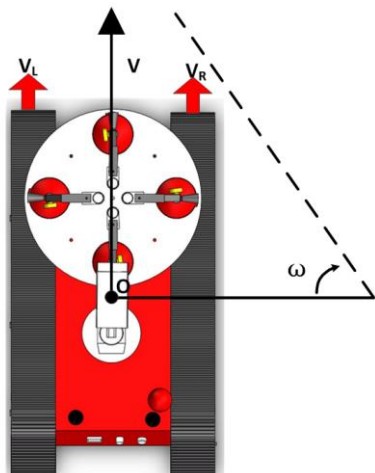

**Figure 9.** Motion model of crawler robot.

Depending on the track speed, the robot can be classified into four motion states: If the track speed on both sides is the same, the robot's mobility direction is straight ahead or straight behind. Steering motion will be produced if the track speeds on both sides differ, with the track on the side with the slower speed serving as the steering direction. While one side of the track rotates, the other remains stationary. The robot must stop the track's movement at the center to produce in situ rotation. The robot will rotate in place with the beam of the two tracks as the center when the two tracks rotate at the same speed and in opposite directions.

### 3.2. Design of the Robot's Automatic Fire-Extinguishing System

An automatic fire-extinguishing system includes video monitoring and fire-extinguishing systems. Flames are detected and located using a video surveillance system. The fire-extinguishing system aims at the flame based on its location information and sprays an extinguishing agent to put out the fire.

#### 3.2.1. Video Surveillance System

The video surveillance system consists of a thermal imaging camera and a PTZ (Pan/Tilt/Zoom). A Dahua DH-TPC-BF5400 thermal imaging camera with clear and reliable imaging was used in this study. The camera is mounted on the PTS-303Z cradle to extend the monitoring range, and the thermal camera is connected to the industrial computer via network connections. Figure 10 shows the video surveillance system.

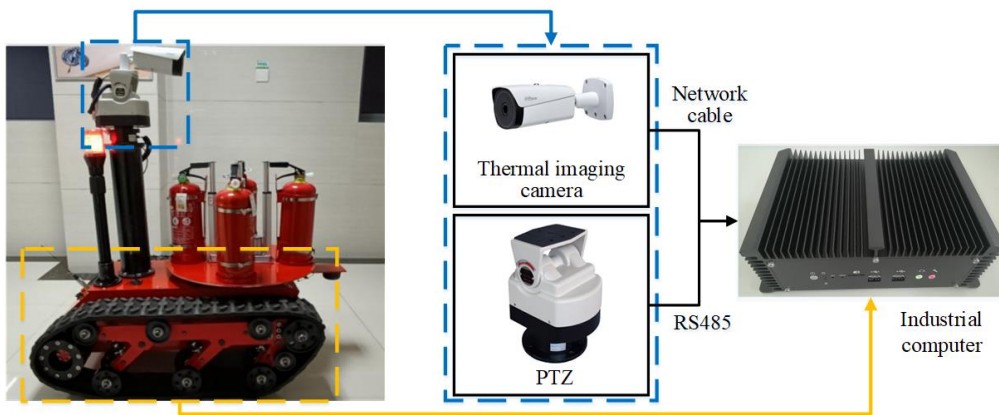

**Figure 10.** Video surveillance system.

The thermal imaging camera's video image data are fed into the flame detection model for detection. If a flame is detected, the distance between the flame center coordinate and the image center is calculated; if a flame is not detected, the flame detection continues. The positive and negative position deviations are judged when the position deviation exceeds the defined pixel value. If the position deviation is negative, the robot will receive a left rotation control instruction. The robot is sent the proper rotation control instruction, and if the position deviation is positive, the flame position deviation is calculated simultaneously. The rotation stops when the position divergence is less than the set pixel value.

### 3.2.2. Fire-Extinguishing System

Due to the inability of the current firefighting robot's fire-extinguishing system to cope with the diversity of the environment, this paper proposes a rotating fire-extinguishing system, as illustrated in Figure 11. The robot's fire-extinguishing equipment consists of four portable fire extinguishers. Four fire extinguisher cans are evenly spaced on the robot's disc. Various types of fire extinguishers can be combined and matched to combat various types of fires. Four fire extinguisher canisters can be sprayed alternately or simultaneously by controlling the rotating device and the fire extinguisher trigger.

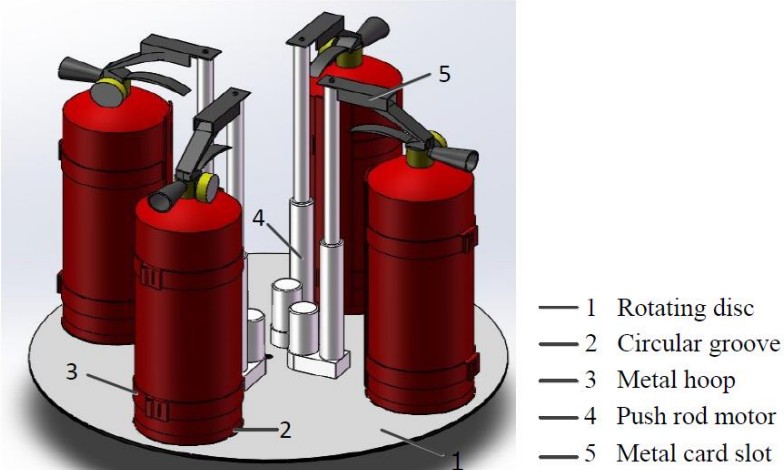

| | |
|---|---|
| 1 | Rotating disc |
| 2 | Circular groove |
| 3 | Metal hoop |
| 4 | Push rod motor |
| 5 | Metal card slot |

**Figure 11.** Fire-extinguishing system installed on the robot.

The fire-extinguishing system comprises the following components: fire extinguisher fixtures, trigger device, rotating device, and system control device.

(1) Fire extinguisher fixtures

The robot's fire extinguisher fixtures can install and replace various fire extinguishers. The fire extinguishers are inserted into four circular grooves in the rotating disc of the robot. Two metal clasps are positioned above the grooves to secure the fire extinguishers further. Screws can be used to adjust the clasps' tightness.

(2) Fire extinguisher trigger device

The fire extinguisher trigger device is adjacent to the fire extinguisher fastening frame. The fire extinguisher trigger device comprises a pushrod motor and a concave metal card slot. The concave metal card slot is mounted on the top of the pushrod motor, which is attached to the rotating disc via a pushrod. The pushrod motor descends when the fire extinguisher must be triggered, driving the concave metal card slot to press the fire extinguisher handle.

(3) Rotating device

The motor's rotation drives the revolving disc's rotation. The rotating disc is secured to the rotating bearing, secured to the robot chassis via the rotating bearing base. A giant gear is installed on the outside of the rotating bearing, and a small gear is installed on the top of the motor's rotating shaft. A DC motor powers the biting between the large and small gears.

To ensure precise fire extinguishing, it is necessary to ensure that the fire extinguisher rotates precisely to the front of the robot when the rotating disc rotates. In this article, the revolving disc is made of aluminum. A 1004NO NPN proximity switch sensor that is only sensitive to iron is installed below it, along with a metal foil attached to the underside of each fire extinguisher fixture. Figure 12 illustrates the proximity switch's installation position schematically. When the proximity switch detects iron, the robot's fire extinguisher nozzle is directly ahead. At this point, the proximity switch signal is received by the fire-extinguishing system's control device, which immediately issues the stop command.

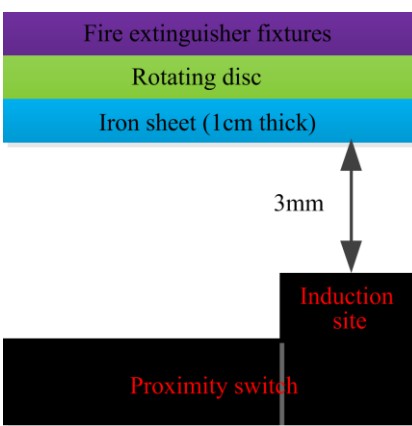

**Figure 12.** Schematic diagram of the installation position of the proximity switch.

(4) System control device

In this paper, the STM32F103ZET6 microcontroller serves as the controller for the automatic fire-extinguishing system. To drive and control the rotating device's four pushrod motors and DC motors, the L289N DC motor drive module is selected. The L289N is capable of driving and controlling two direct-current motors simultaneously. As a result, this system employs three L289Ns to control the drive of five DC motors.

## 4. ROS-Based Map-Building System and Automatic Fire-Extinguishing System Implementation

ROS is a software framework for developing robots that supports various programming languages and includes the Gazebo simulation platform and the RVIZ visualization

tool, enabling programmers to more easily design and debug mobile robot-related applications [40]. ROS was installed on Ubuntu 16.04 in this study, and its corresponding version was Kinetic. A fire inspection robot's software system is composed primarily of a remote control system, an industrial computer software system, and a bottom-drive control system. Through ROS distributed multi-machine communication, the computer running the remote control system communicates with the industrial computer. This work transplanted the map-building and flame detection algorithms into the ROS system as nodes to increase the stability of the system's programs, allowing all programs to perform unified engineering management and data communication within the ROS system.

### 4.1. Map-Building System Implementation

Figure 13 depicts the Cartographer map construction system's node diagram. The majority of the robot map construction system comprises multiple ROS nodes; /mbot bringup is the robot's bottom control node, and the topic is /firerobot odom wheel odometer; /rplidarNode is a lidar node, and the topic is scan—that is, Lidar cloud point information; /imu is the node for the IMU sensor, and the topic is /imu/data raw; /base link to imu link is a node for static coordinate conversion, and the topic is/tf coordinate conversion, which performs the static conversion between the robot's base coordinate base link and the IMU coordinate. The extended Kalman filter node/robot pose ekf subscribes to the imu and odom nodes to perform information fusion and then publish their coordinate information via the /tf coordinate change.

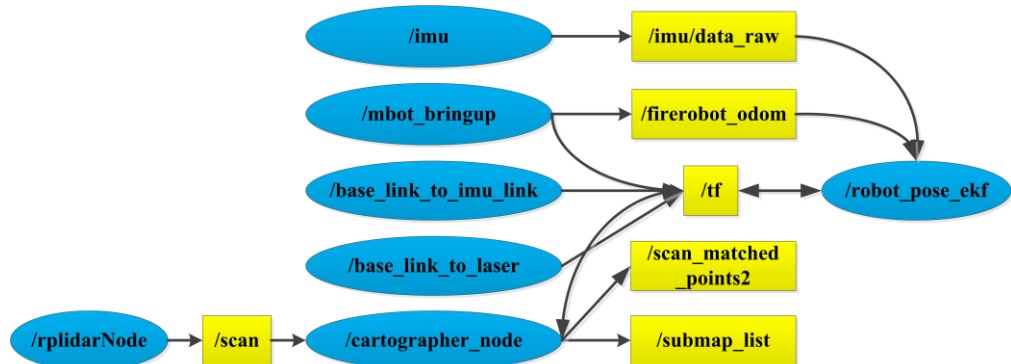

**Figure 13.** ROS node diagram of the Cartographer map construction system.

### 4.2. Realization of the Automatic Fire-Extinguishing System

An ROS node diagram of an automatic fire-extinguishing system is shown in Figure 14. At the far left of the illustration is the image acquisition node /ThermalImagingCamera of a thermal imaging camera, which is responsible for capturing video images; /image listener is the image preprocessing node, which subscribes to the topic /ThermalImagingCamera compressed published by /ThermalImagingCamera and preprocesses the video image to complete the image format conversion; /darknet ros is a flame detection model node used for flame detection, which processes the flame detection results and the flame coordinate information and publishes the /darknet ros/bounding boxes topic; /fire aim is the flame aiming node. The flame aiming function is accomplished by controlling the robot's rotation based on the flame coordinate information.



**Figure 14.** ROS node diagram of the automatic fire-extinguishing system.

## 5. Results and Analysis

### 5.1. Building Fire Simulation Experimental Device Setup

In this study, a building fire simulation experimental device, as shown in Figure 15a, was constructed to evaluate the autonomous inspection and automatic fire-extinguishing functions of robots in the fire environment. The experimental device as a whole is an "L"-shaped corridor. A simulated fire field experimental environment with a length of 10 m, a width of 1.5 m, and a height of 2.2 m is formed inside the device. The external console and the device's display screen can control and display the experimental equipment inside the device. The dry powder fire extinguisher agent employed in the experiment is corrosive to some extent. To protect the experimental device in the experiment, a glass magnesium fireproof board with a length of 2 m and a width of 1.2 m is placed on both sides of the burning area, and a glass magnesium fireproof board is also placed in the internal test, as shown in Figure 15b.

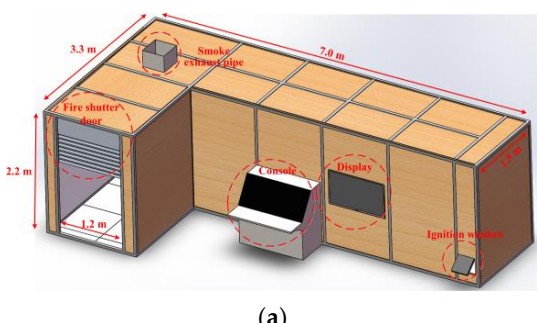 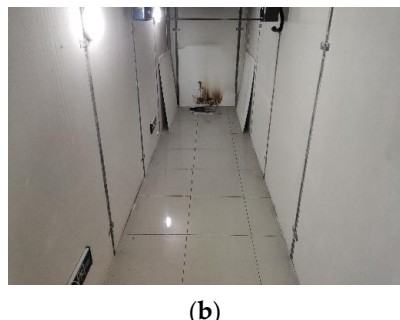

(**a**)                                                   (**b**)

**Figure 15.** Experimental setup: (**a**) Building fire simulation experimental device. (**b**) Inside the device before the experiment.

### 5.2. Robot Map Construction

The cotton rope was lit during the experiment to create a fire environment. Figure 16 depicts the cotton rope used in the experiment. In the robot, the distance between the laser radar and the innermost wall of the corridor is 3.8 m. The experimental procedure is depicted in Figure 17. On the left is a picture of the experimental environment taken by the camera embedded in the experimental device. On the right is a laser point cloud generated by Lidar scanning. A dot in a laser point cloud represents a laser beam, i.e., a ranging point.

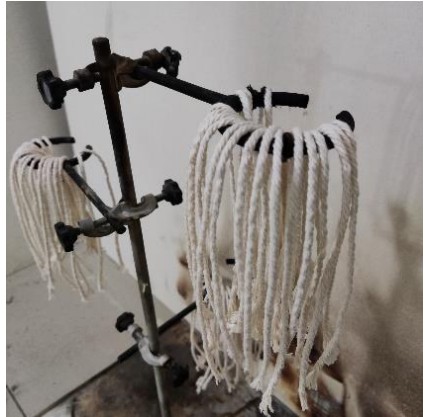

**Figure 16.** Cotton rope to be ignited.

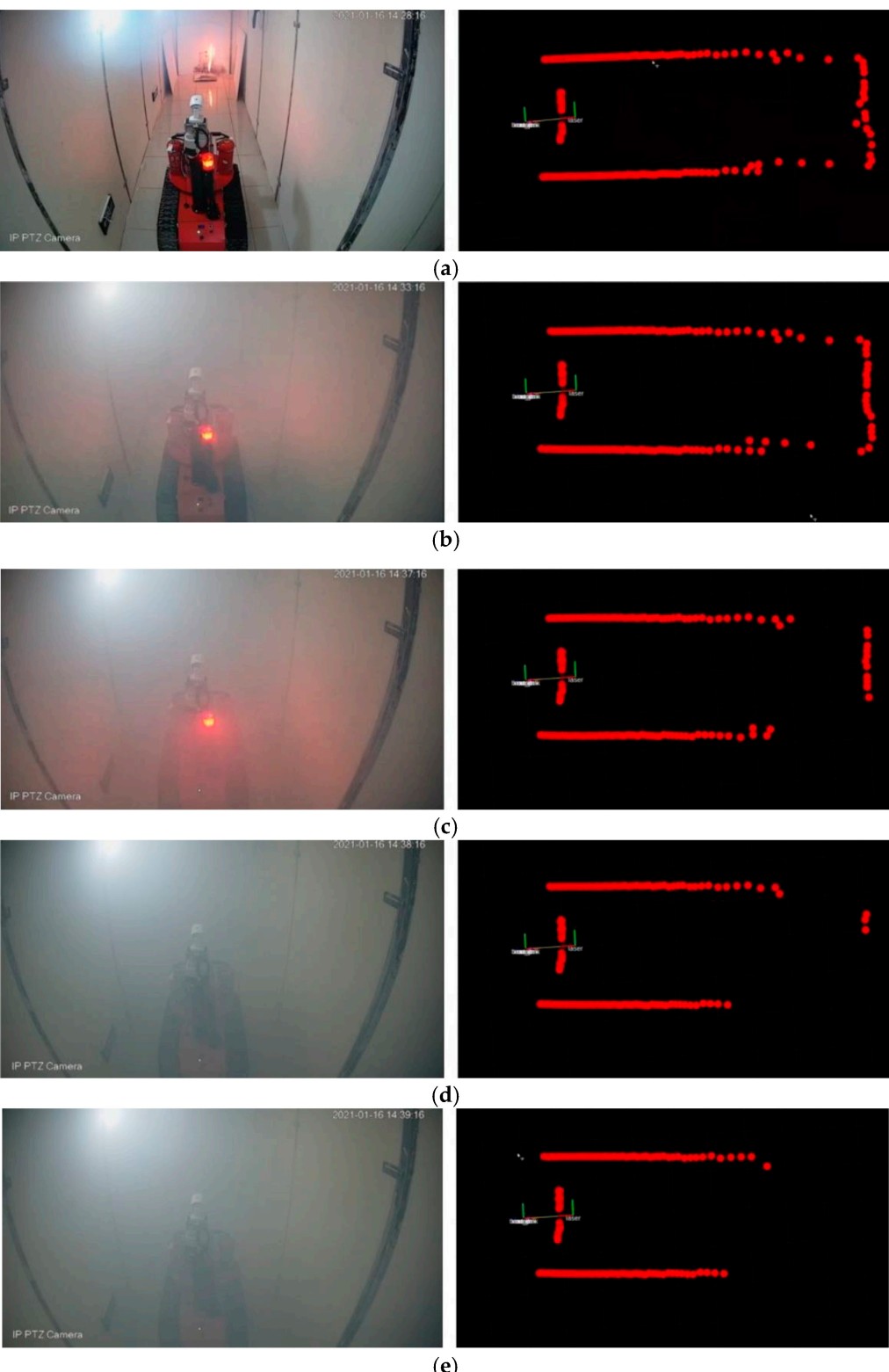

**Figure 17.** Laser point cloud in the fire environment: (**a**) Freshly lit cotton rope. (**b**) Cotton rope lit for 5 min. (**c**) Cotton rope lit for 9 min. (**d**) Cotton rope lit for 10 min. (**e**) Cotton rope lit for 11 min.

As illustrated in Figure 17, smoke and other factors at the fire site interfered with the Lidar ranging five minutes after the cotton rope was lit. At 9 min, the number of laser beams on the far right of the laser point cloud decreased. After 10 min, only three laser beams remained on the laser point cloud's far right. At 11 min, the laser beam to the right

of the laser point cloud vanished completely. The experimental results indicate that the fire environment affects the maximum measuring distance of Lidar. However, it can continue to operate normally, except that the laser beam at the farthest point is significantly affected, while the laser beams at other locations remain normal. At this point, the robot is instructed to advance by 1 m, as illustrated in Figure 18.

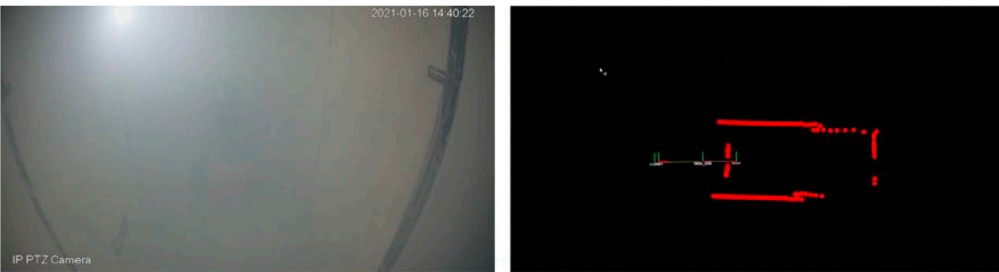

**Figure 18.** The laser point cloud of controlling the robot to move forward 1 m.

As illustrated in Figure 18, the laser beam on the far right reappeared after the machine moved forward by 1 m. The distance between the Lidar and the experimental device's innermost wall at this point was 2.8 m, indicating that the Lidar can detect typical detection barriers within a range of 5.6 m in a fire environment. At this point, the control robot creates a map of the environment, as illustrated in Figure 19.

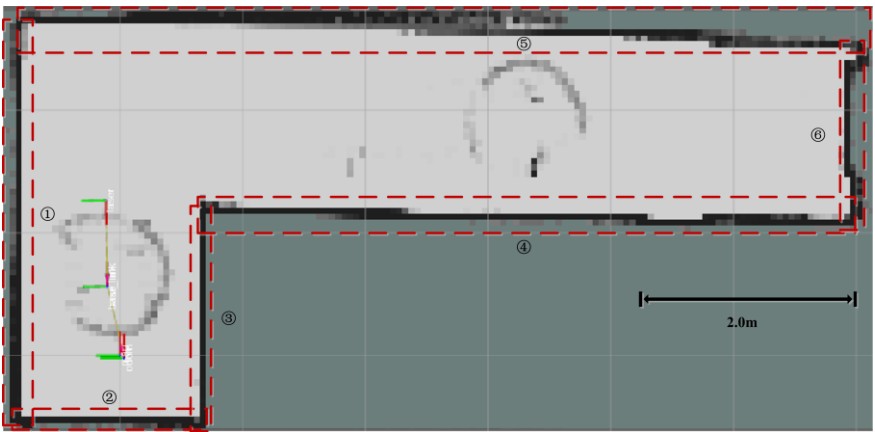

**Figure 19.** The environment map constructed by the robot in the fire environment.

Figure 19 illustrates that the constructed map is comprehensive and has smooth edges. Due to the location of the fire prevention board, the map corresponding to the inner side of the experimental device is slightly frayed. Six locations marked in Figure 19 were chosen for measurement to determine the map's accuracy. The six measurement locations correspond to the six edges of the map that the robot constructed in this paper. The error data for the six measurement positions are shown in Table 2, and the error percentages for the six measurement positions are shown in Figure 20.

**Table 2.** Six edge errors of the map constructed by the robot in the experiment.

| Measuring Position | Actual Length (cm) | Map Build Length (cm) | Absolute Error (cm) | Error Percentage (%) |
| --- | --- | --- | --- | --- |
| 1 | 330.00 | 326.20 | 3.80 | 1.15 |
| 2 | 150.00 | 151.50 | 1.50 | 1.00 |
| 3 | 180.00 | 178.10 | 1.90 | 1.06 |
| 4 | 550.00 | 541.20 | 8.80 | 1.60 |
| 5 | 700.00 | 688.20 | 11.80 | 1.69 |
| 6 | 150.00 | 147.80 | 2.20 | 1.47 |

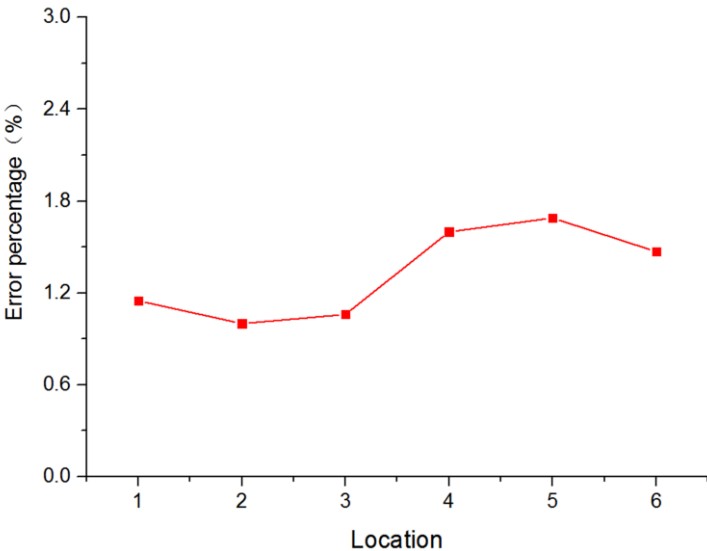

**Figure 20.** Error percentage of measured positions.

The maximum error percentage for each of the six measurement locations was 1.69%, the minimum was 1.00%, and the average was 1.33%. Additionally, as shown in Table 2 and Figure 20, the errors at positions 4, 5, and 6 were slightly greater due to the inclusion of the fire prevention board in the experimental device. The experimental results show that the overall effect is insignificant, although the fire environment interferes with the laser ranging. The robot designed in this paper has a high degree of accuracy in mapping the fire scene, which provides a solid foundation for the robot's automatic inspection and fire extinguishing.

*5.3. Robot Flame Detection and Automatic Aiming*

The flame detection model for the firefighting robot uses an image dataset that was self-constructed in our lab. Firstly, the experimental setup was built to obtain thermal imaging flame images by experimental means and select flame images with high quality; secondly, the flame targets in the images were labeled; finally, the training and validation sets were divided to construct a standard dataset. The dataset contained a total of 14,757 thermal infrared flame images, of which 70% were used as the training set and 30% were used as the test set. The training environment parameters for the flame detection model are shown in Table 3.

**Table 3.** Training environment parameters for flame detection model.

| Hardware Environment | Software Environment | Programming Language |
|---|---|---|
| NVIDIA Quadro K2000 4 G | Operating system Ubuntu 16.04, deep learning framework CSPDarknet53 | Python, C language |

The definitions and settings of the major experimental parameters in YOLOv4 are shown in Table 4. Training was performed using pretrained weight files of the official YOLOv4 dataset for migration learning to accelerate convergence and improve learning speed.

**Table 4.** Definition and setting of the experimental parameters.

| Parameter | Parameter Definition | Parameter Value |
| --- | --- | --- |
| max_batches | Maximum number of iterations | 4000 |
| batch | The number of samples participating in a batch of training | 64 |
| subdivision | Number of sub-batches | 16 |
| momentum | Momentum parameter in gradient descent | 0.9 |
| decay | Weight attenuation regular term coefficient | 0.0005 |
| learning_rate | Learning rate | 0.001 |
| ignore_thresh | The threshold size of the IOU involved in the calculation | 0.5 |

The flame detection model was trained and then tested for its ability to detect flames with automatic flame aiming. As shown in Figure 21, the oil pool was placed on a mobile platform with wheels, and the oil pool could be moved by pulling the iron bar at the window. The oil pool was filled with the fuel n-heptane, and the oil pool was 4.5 m away from the flame. The robot flame detection process and automatic flame aiming process are shown in Figure 22.

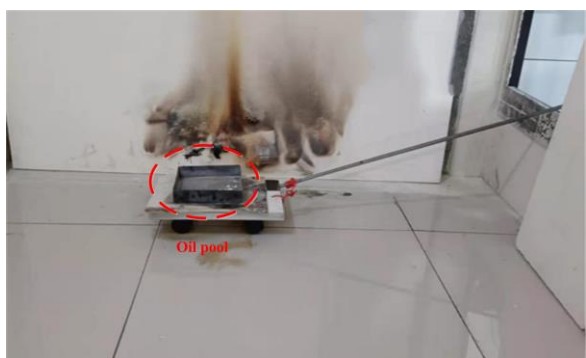

**Figure 21.** Mobile platform and oil pool.

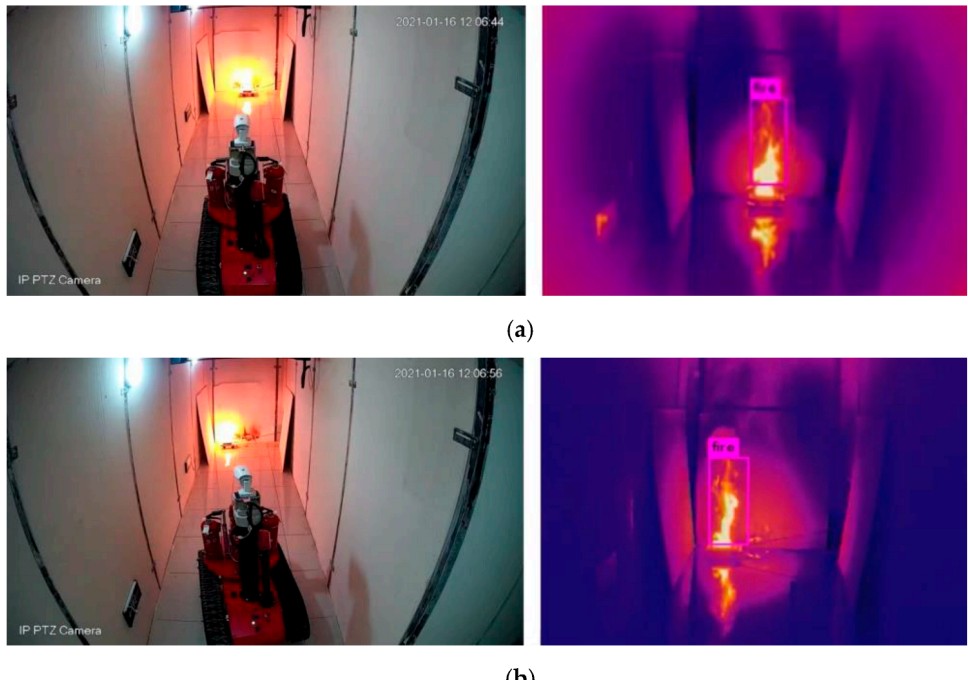

**Figure 22.** The robot can still aim at the flame after moving the flame: (**a**) The robot is aimed at the flame. (**b**) The mobile flame robot turns to automatically aim at the flame.

When the flame is stationary, the robot can automatically identify the flame and aim at the flame; if the mobile platform moves, the rotating robot can still identify the flame and automatically aim at the flame. The flame detection confidence of the robot is 72%, and the frame rate is 11 fps, which is displayed at this time on the flame identification terminal of the fire-extinguishing system of the firefighting robot. The frame rate meets the requirement of real-time processing.

### 5.4. Robot Autonomous Inspection and Automatic Fire Extinguishing

The experiment involved autonomous robot inspection and automatic fire extinguishing. The robot plans its path using the map created in Section 5.2 and the move_base function package in ROS [41]. Cotton rope and cardboard boxes were used as combustion materials in the experiment, and four 3 kg portable dry powder fire extinguishers were used as extinguishing agents. The burning materials and fire extinguishers used in this experiment are shown in Figure 23.

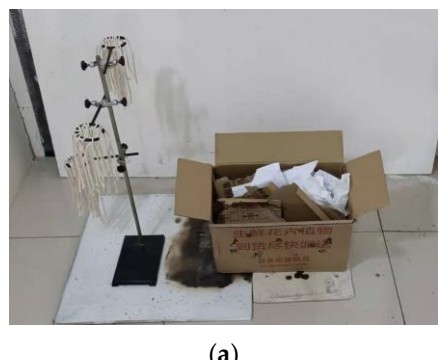
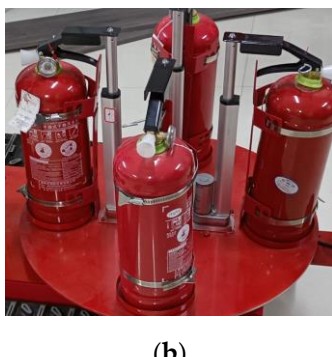

(**a**)　　　　　　　　　　　　　　　(**b**)

**Figure 23.** Burning materials and fire extinguishers carried by robot: (**a**) Burning materials. (**b**) The robot carries four 3 kg dry powder fire extinguishers.

During the experiment, a large amount of smoke was generated using a shaded combustion cotton rope fire to rapidly reduce the experimental device's visibility, and the cardboard box was ignited. The entrance of the experimental device was set as the starting point of the robot inspection, the built environment map was loaded, the inspection location was set, and the path planning was carried out using the move_base function package of ROS to achieve independent inspection. The inspection path of the robot is shown in Figure 24. The robot detects the flame during the inspection process and automatically extinguishes the fire, and the robot's automatic fire-extinguishing process is shown in Figure 25.

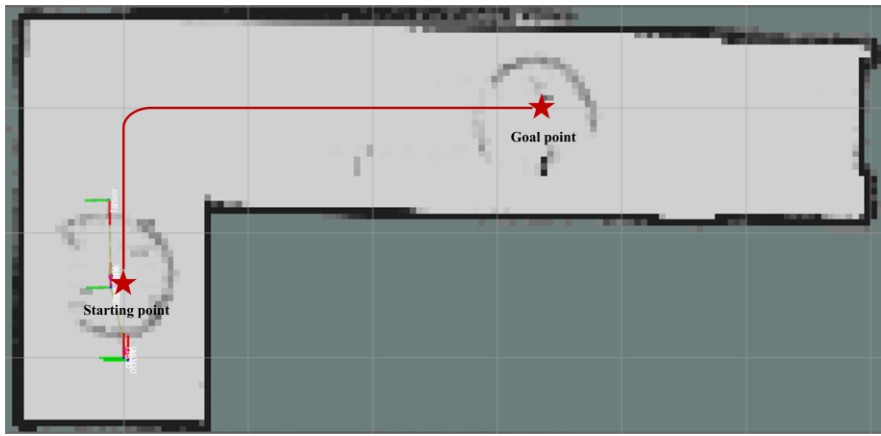

**Figure 24.** Inspection path of the robot.

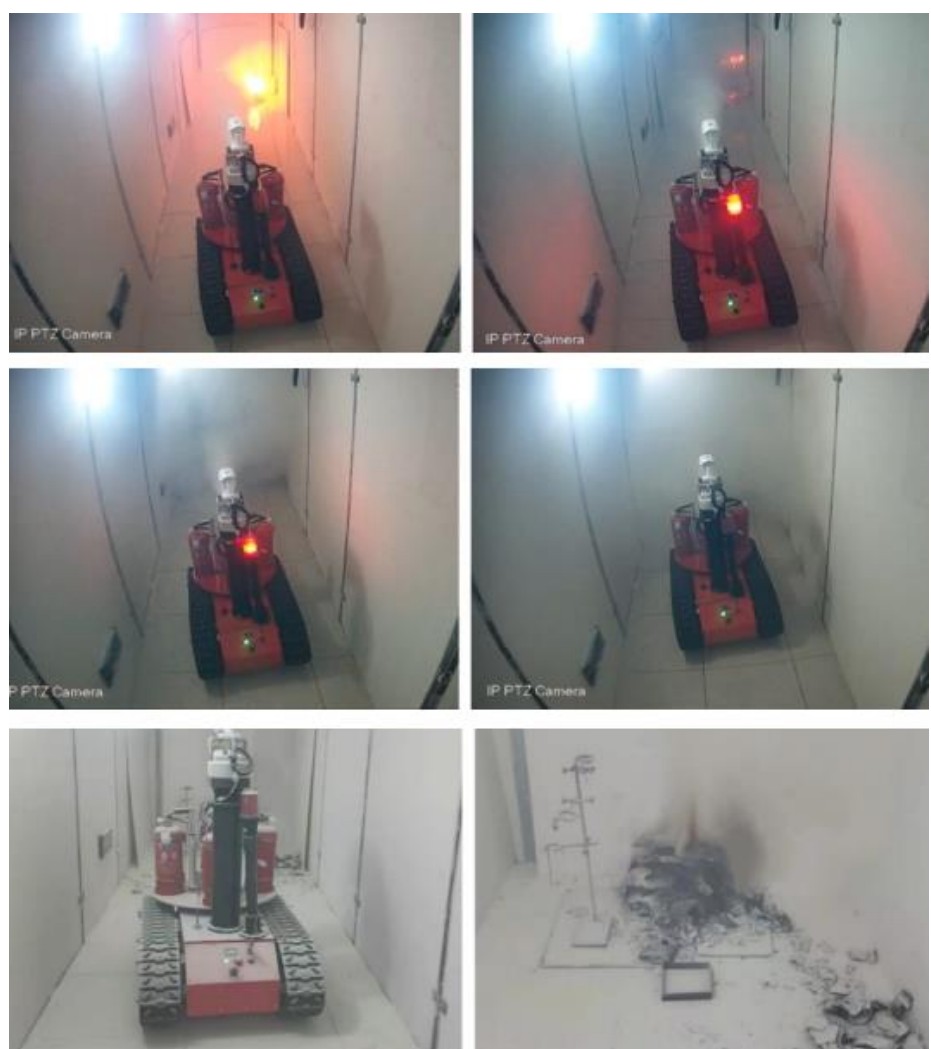

**Figure 25.** The robot finds flames and extinguishes them during the autonomous inspection.

## 6. Conclusions

This study developed a fire inspection robot with two parts: autonomous inspection, and automatic fire extinguishing. The hardware for the autonomous inspection system primarily consists of a motion unit and sensor unit. In contrast, the hardware of the automatic fire-extinguishing system primarily consists of a video monitoring system and a fire-extinguishing system. The Cartographer map construction algorithm is used to create a real-time map of the fire scene environment, and YOLOv4 deep-learning-based flame detection technology is used to detect flames. We constructed an L-shaped corridor fire environment simulation experiment device to evaluate the robot's performance. The experimental device lit a cotton rope to create a fire environment. When the cotton rope was burned, a portion of the laser point cloud near the fire vanished; when the robot advanced 1 m, the point cloud reappeared, indicating that the fire environment did not affect the robot's map building. By analyzing the constructed map, we can determine that the average error percentage of map construction accuracy is only 1.33%, providing a solid foundation for autonomous robot inspection and fire extinguishing. Subsequently, in the robot's autonomous inspection and automatic fire-extinguishing experiment, the robot was able to detect and extinguish the flame in real time during the autonomous inspection process.

The robot performs two functions: autonomous inspection, and automatic fire extinguishing, reducing firefighters' workload and decreasing the frequency of significant fires.

Limited by the problem of laboratory space and the danger of fire experiments, the current fire simulation experimental device is small. In the following work, we will actively work with relevant companies to conduct further testing on the usage performance of the robot through industry–academia integration.

**Author Contributions:** Conceptualization, S.L.; methodology, S.L., J.Y. (Junying Yun) and C.F.; software, J.Y. (Junying Yun) and C.F.; validation, S.L., J.Y. (Junying Yun), C.F. and G.S.; formal analysis, D.Z.; investigation, Y.G. and J.Y. (Jialuo Yang); writing—original draft preparation, S.L. and J.Y. (Junying Yun); writing—review and editing, S.L. and J.Y. (Junying Yun). All authors have read and agreed to the published version of the manuscript.

**Funding:** This work was supported by the National Natural Science Foundation for Young Scientists of China (Grant No. 51804278), the Training Plan for Young Backbone Teachers in Colleges and Universities in Henan Province (Grant No. 2021GGJS094), the Henan Science and Technology Development Plan (Grant No. 212102210535), and the National Innovation Training Program for College Students (Grant No. 202210462007).

**Institutional Review Board Statement:** Not applicable.

**Informed Consent Statement:** Not applicable.

**Data Availability Statement:** Not applicable.

**Conflicts of Interest:** The authors declare no conflict of interest.

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
