# Peer review of "An Indoor Autonomous Inspection and Firefighting Robot Based on SLAM and Flame Image Recognition"

_fire, doi:10.3390/fire6030093_

Round 1
Reviewer 1 Report
the motivation must be clear in the abstract. It's not clear how firefighting robot is an alternative to fire alarms.
The abstract emphasizes on indoor but the introduction focusses on the general fire incidents (non-indoor). The motivation for the introduction needs re-write.
Shorten and re-write the following the sentence to clarify the motivation clearly. Best to break it down into smaller sentences:
currently unsuitable for use in an indoor environment, as evidenced by the following factors: Current firefighting robots are bulky and inconvenienced. Currently, the firefighting robot uses water as the primary extinguishing medium and requires an external water belt, but the indoor space is small, there are many turns, and the water belt restricts the robot’s movement; Water as a fire extinguishing medium ....
A lot of sentences need to be re-written, and they require grammatical check. For example:
This statement needs more explanation. What is meant by "not enough"?: The current intelligent level of the firefighting robot is not enough....
Re-write this sentence -> traditional image detection is a kind of detection system using rules.
not clear what this means -> by manually designed feature threshold.
Since you consider deep learning flame detection as a contribution, please provide more detailed explanation of the YOLOv4 implementation. What pre-processing techniques have conducted and how do you feed in the flame features into yolo. What features do you feed in exactly. Explain how you develop the flame detection model? What YOLOv4 parameters do you use?
The contribution of the paper is not clear. Is it a new firefighting robot using laser or the deep learning flame detection.
If the contribution is robot designed user laser SLAM, what differentiates this work with the following: Liu, Y. T., Sun, R. Z., Zhang, X. N., Li, L., & Shi, G. Q. (2021). An autonomous positioning method for fire robots with multi-source sensors. Wireless Networks, 1-13.
If the contribution is due to an automatic rotary fire extinguishing device, what differentiates this work with:
Zhang, S., Yao, J., Wang, R., Liu, Z., Ma, C., Wang, Y., & Zhao, Y. (2022). Design of intelligent fire-fighting robot based on multi-sensor fusion and experimental study on fire scene patrol. Robotics and Autonomous Systems, 154, 104122.
The performance of the firefighting robot needs more quantitatively analysis. The results in Table 2 is not sufficient as it only analyses 6 measure location without solid statistical robustness.
Author Response
- The motivation must be clear in the abstract. It's not clear how firefighting robot is an alternative to fire alarms.
Response 1. The firefighting robot is mobile, and better in initiative than the fire alarms. In addition, the firefighting robot can be equipped with the automatic fire extinguishing system, which enables it to achieve precise fire extinguishing once a fire is detected. In the revised manuscript, the authors consider that the description of fire alarms is irrelevant to the article and have removed it. Please see the revised abstract for more details in P1.
2.The abstract emphasizes on indoor but the introduction focusses on the general fire incidents (non-indoor). The motivation for the introduction needs re-write.
Response 2. Thank you for your valuable comments. The introduction section of the manuscript has been reworked. We have made the following changes in the revised manuscript: 1) an indoor fire example to emphasize the hazards of indoor fires is added; 2) the reason of why current firefighting robots are not suitable for indoor environments is elaborated; and 3) the functions of the robot we designed is streamlined. For more details, please see P1~4 in the revised manuscript.
- Shorten and re-write the following the sentence to clarify the motivation clearly. Best to break it down into smaller sentences:
currently unsuitable for use in an indoor environment, as evidenced by the following factors: Current firefighting robots are bulky and inconvenienced. Currently, the firefighting robot uses water as the primary extinguishing medium and requires an external water belt, but the indoor space is small, there are many turns, and the water belt restricts the robot’s movement; Water as a fire extinguishing medium ....
A lot of sentences need to be re-written, and they require grammatical check. For example:
This statement needs more explanation. What is meant by "not enough"?: The current intelligent level of the firefighting robot is not enough....
Re-write this sentence -> traditional image detection is a kind of detection system using rules.
not clear what this means -> by manually designed feature threshold.
Response 3. Thank you for your valuable comments. We have rewritten the relevant sentences to clarify the meaning what we want to express. And also, the grammar of the whole article has been rechecked.
- Since you consider deep learning flame detection as a contribution, please provide more detailed explanation of the YOLOv4 implementation. What pre-processing techniques have conducted and how do you feed in the flame features into yolo. What features do you feed in exactly. Explain how you develop the flame detection model? What YOLOv4 parameters do you use?
Response 4. Thank you very much for your suggestion. This paper focuses on the development of the firefighting robot as a whole and its functionality. This firefighting robot uses the YOLOv4 algorithm as the flame detection algorithm. The YOLOv4 training process include: 1) construct the infrared image flame dataset based on the existing official dataset standard; 2) set the main training parameters of the model, which the values we use are shown in Table 1; 3) the model training adopts the pre-training weight file of YOLOv4 official data set for migration learning, to accelerate convergence and improve learning speed; 4) Test validation of the trained weights is performed using the test set. For more details, please refer to P17~19 of the revised manuscript.
Table 1 Training parameters configuration
|
max_batches |
batch |
subdivision |
momentum |
|
4000 |
64 |
16 |
0.9 |
|
decay |
learning_rate |
ignore_thresh |
|
|
0.0005 |
0.001 |
0.5 |
|
- The contribution of the paper is not clear. Is it a new firefighting robot using laser or the deep learning flame detection.
If the contribution is robot designed user laser SLAM, what differentiates this work with the following: Liu, Y. T., Sun, R. Z., Zhang, X. N., Li, L., & Shi, G. Q. (2021). An autonomous positioning method for fire robots with multi-source sensors. Wireless Networks, 1-13.
If the contribution is due to an automatic rotary fire extinguishing device, what differentiates this work with: Zhang, S., Yao, J., Wang, R., Liu, Z., Ma, C., Wang, Y., & Zhao, Y. (2022). Design of intelligent fire-fighting robot based on multi-sensor fusion and experimental study on fire scene patrol. Robotics and Autonomous Systems, 154, 104122.
Response 5. This paper systematic designs a new firefighting robot with the abilities of automatic inspection, automatic flame detection, and automatic fire extinguishing. And fully consider the connectivity between the complex hardware, and the applicability of the algorithm in the fire environment.
The paper "An autonomous positioning method for fire robots with multi-source sensors" also use SLAM technology for positioning. While, they do not design other systems, like the flame detection system and the fire extinguishing system which are very important for the firefighting robot.
In the article "Design of intelligent fire-fighting robot based on multi-sensor fusion and experimental study on fire scene patrol," the firefighting robot's extinguishing medium is water. While, there are many various combustion materials in the indoor environment, some cannot use water to extinguish. Thus, our paper uses the fire extinguishers to design the extinguishing system. And the user can install four different extinguishers as required. The robot can select the appropriate fire extinguishers and spray them automatically.
- The performance of the firefighting robot needs more quantitatively analysis. The results in Table 2 is not sufficient as it only analyses 6 measure location without solid statistical robustness.
Response 6. Thank you for your valuable comments! Due to the size of our building fire simulation experimental setup, the size of the constructed maps is limited, so only the locations of 6 measurement points are analyzed in this paper. In the next step, we will expand the building fire simulation experimental setup to make the statistics more reliable.

Reviewer 2 Report
Is Figure 23 appropriate in the paper?
How is the dataset used for YoloV4 algorithm training constructed?
The article points out that traditional automatic fire alarm systems have a slow response time. How to prove the timeliness of the author's algorithm for flame detection?
Author Response
- Is Figure 23 appropriate in the paper?
Response 1. Thank you very much for your question. Figure 23 shows the firefighting robot's terminal interface to identify the flame in the flame detection experiment. The purpose of inserting Figure 23 in the manuscript is to allow the reader to see the experimental results visually.
After revisiting the previous version of the manuscript, we found that Figure 23 was slightly redundant. Thus, Figure 23 has been removed in the revised manuscript, and the corresponding text section has been modified accordingly. For more details, please refer to P18~19 of the revised manuscript.
- How is the dataset used for YoloV4 algorithm training constructed?
Response 2. The dataset used for training the Yolov4 algorithm is self-constructed by our lab, and has been described in lines 469~475 of P17 in the revised manuscript. We have applied for a GitHub account: wangyeheng123, and we will upload the dataset to the GitHub account when the finishing work is completed.
- The article points out that traditional automatic fire alarm systems have a slow response time. How to prove the timeliness of the author's algorithm for flame detection?
Response 3. The firefighting robot is mobile, and better in initiative than the traditional fire alarm system. In the flame detection experiments of the firefighting robot, the flame detection algorithm is processed at a frame rate of 11 fps, which satisfies the real-time processing requirements.
